# Synthesis of Spherical TiO$_2$ Particles with Disordered Rutile Surface for Photocatalytic Hydrogen Production

**Na Yeon Kim, Hyeon Kyeong Lee, Jong Tae Moon and Ji Bong Joo \***

School of Chemical Engineering, Konkuk University, Gwangjin-gu, Seoul 05029, Korea;
kny960403@konkuk.ac.kr (N.Y.K.); hyeonk@konkuk.ac.kr (H.K.L.); aidf91@gmail.com (J.T.M.)
\* Correspondence: jbjoo@konkuk.ac.kr; Tel.: +82-245-03-545

**Abstract:** One of the most important issues in photocatalysis research has been the development of TiO$_2$-based photocatalysts that work efficiently under visible light conditions. Here, we report the monodispersed, spherical TiO$_2$ particles with disordered rutile surface for use as visible-light photocatalysts. The spherical TiO$_2$ particles with disordered surface were synthesized by sol-gel synthesis, followed by sequential calcination, and chemical reduction process using Li/Ethylenediamine (Li/EDA) solution. Variation of the calcination temperature allowed the crystalline properties of the calcined TiO$_2$ samples, such as the ratio of anatase and rutile, to be finely controlled. The content ratios of anatase phase to rutile phase leads to different degrees of disorder of the rutile surface, which is closely related to the photocatalysis activity. Chemical reduction using the Li/EDA solution enables selective reduction of the rutile surface of the calcined TiO$_2$, resulting in enhanced light absorption. As a result, we were able to synthesize spherical TiO$_2$ photocatalysts having a disordered rutile surface in a mixed crystalline phase, which is beneficial during photocatalysis in terms of light absorption and charge separation. When used as photocatalysts for hydrogen production under solar light conditions, the chemically-reduced TiO$_2$ particles with both the disordered rutile surface and mixed crystalline phase showed significantly enhanced catalytic activity.

**Keywords:** TiO$_2$; spherical particle; disordered surface; photocatalysts; hydrogen production

---

## 1. Introduction

In modern society the energy crisis is becoming one of the biggest issues to directly impact our lives. Hydrogen as a green energy carrier has attracted much attention, due to its high energy capacity, environment-friendly characteristics, and sustainability [1]. As people are recognizing that high concentration of carbon dioxide is closely related to global warming and climate change, hydrogen can be considered as a one of the representative alternative-energy resources to either reduce or replace the use of depletable fossil fuels. There are several strategies to produce hydrogen, such as the reforming of either fossil fuel or renewable biomass, water electrolysis, ammonia decomposition, photo-electrochemical water splitting, and photocatalytic water splitting [2–8]. Among them, photocatalytic hydrogen production from water is considered as an ideal and economically-feasible method, since infinite solar energy can be used with any other type of energy resource [9].

Titanium dioxide (Titania, TiO$_2$) is one of the most well-known semiconductor photocatalysts. TiO$_2$ materials have a few advantages, which include low cost, considerable photocatalytic activity, low toxicity, high chemical stability, and abundance on Earth [10–12]. Since Honda and Fujishima first discovered hydrogen production by the photoelectrochemical splitting of water under UV light conditions [13], TiO$_2$ has not only been intensively studied with fundamental researches, but also widely

used for practical systems for solar energy conversion [14–17]. Although $TiO_2$ has been intensively studied over the past decades and has become known as a superb photocatalyst, the practical use of $TiO_2$ has still been limited, due to its intrinsic optical property. Since $TiO_2$ has a wide band-gap energy of 3.0–3.2 eV, its photocatalytic performance is limited to the ultra-violet (UV) region [12,17,18]. Even though the greater portion of solar light is in the visible and IR region, $TiO_2$ can absorb mainly UV light, resulting in it generally showing low solar-to-chemical efficiency under solar light conditions. In addition, rapid recombination of photogenerated electron-hole pairs also leads to low quantum efficiency. Thus, this results in low overall photocatalytic activity [19,20].

In order to overcome the above drawbacks, various novel approaches have been taken to improve its optical, electronic, and chemical properties. To narrow the band-gap, either nonmetal or metal ions are doped into $TiO_2$ crystal lattice, regulating either the level of conduction band or valence bands [21–27]. Surface sensitization using organic dyes can allow $TiO_2$ to utilize the exited electron from dye molecules under visible light irradiation [28,29]. Recently, decoration of plasmonic metal nanoparticles on the $TiO_2$ surface have also been suggested to improve hot electron transfer from metal nanoparticles to the conduction band of $TiO_2$, resulting in unexpected photocatalytic performance under visible light conditions [30,31]. Since Mao and co-worker made the pioneering discovery of black $TiO_2$ nanocrystal through surface-disordering using hydrogen [32], there have been many further reports about the interesting strategy of the synthesis of surface-disordered $TiO_2$ by chemical reduction [33–35]. Unlike other $TiO_2$-based photocatalysts by conventional disordering processes, they can achieve significantly disordered surface of $TiO_2$ nanocrystal with well-maintained crystalline property, resulting in high photocatalytic activity on both organic dye decomposition and hydrogen production under solar light [32]. It is also well known that active metal, such as Al and Mg, can reduce $TiO_2$, resulting in the formation of colored $TiO_2$. Huang et al. developed a new approach to prepare colored $TiO_2$ based on Al-reduction [36]. Sinhamahapatra and Yu synthesized black $TiO_2$ by mixing the commercial $TiO_2$ nanocrystals with Mg powder, followed by annealing in the hydrogen environment [37,38]. Park et al. also developed chemical reduction using a Lithium/Ethylenediamine mixture, and prepared blue-colored $TiO_2$ that had the selectively disordered rutile surface. Lithium/Ethylenediamine (Li/EDA) solution, a metal chelate compound, is a very strong reducing agent. Ethylenediamine (EDA) is a superbase chemical that provides high pH conditions. The chelated metallic Li derived from the Li/EDA solution then selectively breaks the bonds between Ti and O of the rutile phase $TiO_2$, resulting in various defects and a disordered surface. Defects in the disordered surface can narrow the band gap, forming interstates between the conduction band and valence band. Therefore, white $TiO_2$ turns into a blueish color [39].

As shown in the previous literature, several chemical reduction approaches can allow the band-gap of pure $TiO_2$ to be narrow, and the colored $TiO_2$ to have high performance under solar light photocatalysis. Although intensive investigation of the synthesis of colored $TiO_2$ was conducted using commercial $TiO_2$ nanocrystals, such as P25 [39–41], there has been only limited study reported on the fabrication of colloidal $TiO_2$ nanostructure. We recently found that uniform colloidal $TiO_2$ particles with tunable crystalline properties, such as the ratio of anatase–rutile phase and crystallinity, can be produced by the sol-gel synthesis of titanium-alkoxide precursors, followed by calcination at different temperatures [42]. As-calcined $TiO_2$ samples show monodispersed colloid particles and well-developed $TiO_2$ crystallinity, with finely-tunable anatase–rutile mixed phase. As previously mentioned, Li/EDA solution can selectively reduce the rutile phase of $TiO_2$, and disorder its surface [39]. Since the crystalline phase of spherical $TiO_2$ particles between anatase and rutile can be finely tuned by varying the calcination temperature while maintaining the morphological dimension, the degree of surface disordering can be systemically controlled, resulting in precise control of the band-gap energy.

In this work, we report the synthesis of the spherical $TiO_2$ particle with the disordered rutile surface for photocatalytic hydrogen production. Specifically, monodispersed $TiO_2$ particles with tunable crystalline property are synthesized by sol-gel synthesis, followed by calcination at different temperatures. Since $TiO_2$ samples have different ratios of anatase to rutile phases, the optical properties

are conveniently controlled by Li/EDA treatment. The resulting Li/EDA-treated $TiO_2$ samples showed advantageous characteristics, such as uniform particle dimension, favorable dispersity, facile absorption of visible light, and controllable degree of disorder of the surface. Corresponding with such reduced $TiO_2$ spherical photocatalysts, it was possible to achieve enhanced performance in photocatalytic hydrogen production under solar light irradiation. We systemically study and discuss the optical properties, physicochemical characteristics, and photocatalytic performance of the spherical $TiO_2$ with disordered rutile surface.

## 2. Results and Discussion

Colored $TiO_2$ spherical particles with the disordered rutile surface were synthesized by a modified sol-gel synthesis, followed by sequential calcination at the desired temperature and chemical reduction, respectively (Figure 1a). More specifically, the synthesis consisted of the following steps: (i) Preparation of a colloidal amorphous $TiO_2$ sphere (AT, amorphous $TiO_2$) (ii) calcination of a $TiO_2$ sphere to convert to the crystalline counterpart (CT-x); and (iii) chemical reduction of crystalline $TiO_2$, to the colored one (RT-X), by using Li/EDA (Li in ethylenediamine) as reducing agent. During preparation of the colloidal $TiO_2$ particle, the monodispersed amorphous $TiO_2$ spheres were synthesized by the sol-gel reaction of titanium n-butoxide (TBOT) in mixed solvent of ethanol and acetonitrile, in the presence of base ammonia and surfactant. The hydrolysis and condensation of the TBOT were highly influenced by several synthetic parameters, such as the solvent environment, the amount of precursor, and the concentration of surfactant. Recently, we systemically studied the effect of the synthetic parameters on the physical–chemical properties of colloidal $TiO_2$ particles, and successfully synthesized uniform $TiO_2$ spheres with controllable crystalline properties [42]. In this work, we also synthesized the uniform spherical $TiO_2$ particles with particle diameter of ca. 290 nm by adapting the previous synthetic method [42]. The SEM image (Figure 1b) clearly shows that uniform spherical particles with a white color powder were well synthesized. After the calcination step, amorphous $TiO_2$ could be crystallized to the crystalline counterparts, which consist of either anatase or rutile phases. The spherical morphology was well maintained, even after high temperature calcination, and the calcined sample showed white color, which is an intrinsic property of crystalline $TiO_2$ (Figure 1c). Chemical reduction using Li/EDA solution could selectively reduce the rutile phase of crystalline $TiO_2$. As previously reported by Park et al., Li-EDA as a strong reducing agent in a superbase can selectively reduce rutile $TiO_2$ to disorder the surface, resulting in black rutile $TiO_2$ having a small band-gap [39]. Since the calcined $TiO_2$ spherical particles had different crystalline ratios of anatase and rutile depending on the calcination temperature, the degree of surface disordering was highly influenced, resulting in the reduced spherical $TiO_2$ particle with different colors. In practical terms, the reduced $TiO_2$ sample showed a blue color with a uniform particle dimension (Figure 1d). After photo-deposition of Pt nanoparticle on the surface of the reduced $TiO_2$ sample, the sample could be used as a photocatalyst for photochemical hydrogen production under solar-light irradiation (Figure 1a).

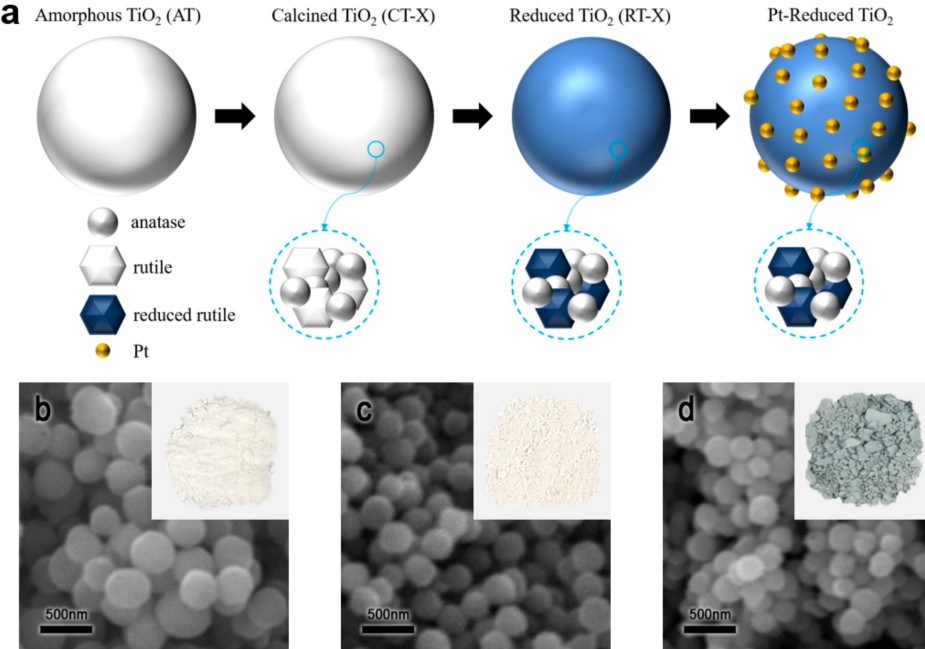

**Figure 1.** (**a**) Schematic illustration for synthesis of reduced TiO$_2$ samples (RT-x) and Pt-reduced TiO$_2$ catalyst. Corresponding SEM and digital images of (**b**) amorphous TiO$_2$ (AT), (**c**) calcined TiO$_2$ (CT-x), and (**d**) reduced TiO$_2$ (RT-x), respectively.

The morphologies of the calcined TiO$_2$ and the reduced TiO$_2$ samples are investigated by SEM. Figure 2a shows the CT-600 (Calcined TiO$_2$ at 600 °C) sample calcined at 600 °C, which reveals uniform spherical morphology with diameter of ca. (281 ± 31) nm. As the calcination temperature increases to 800 °C, the spherical morphology is well maintained, indicating the high thermal stability of the synthesized TiO$_2$ particles (Figure 2b). The average diameter of CT-800 is ca. (280 ± 37) nm, which is almost similar to that of CT-600. Although chemical reduction is carried out to produce the colored TiO$_2$ particles, the overall morphology with diameter is unchanged. All RT-x (Reduced TiO$_2$) samples show the monodispersed, spherical morphology with similar diameter compared to the mother CT sample, indicating high chemical stability (Figure 2c,d).

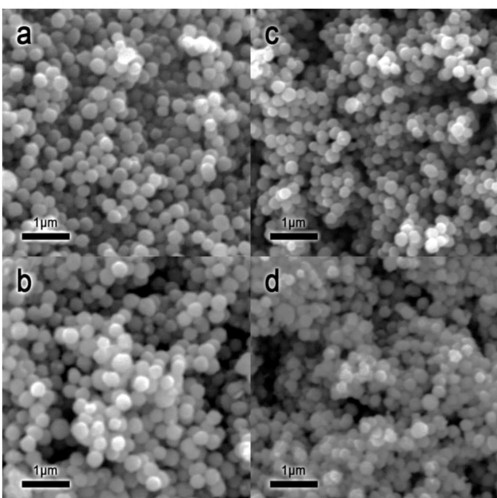

**Figure 2.** SEM images of (**a**) CT-600, (**b**) CT-800, (**c**) RT-600, and (**d**) RT-800. The letters CT and RT indicate calcined and reduced TiO$_2$, respectively. The number after CT and RT indicates its calcination temperature.

The crystalline characteristics of the calcined $TiO_2$ and the reduced $TiO_2$ samples are investigated by X-ray diffraction (XRD). Figure 3a shows the CT-500 sample revealed representative diffraction peaks of $TiO_2$ anatase phase at $2\theta$ = (25.4°, 37.84°, 48.12°, 54.02°, 55.08°, and 62.68°), which are attributed to the (101), (004), (200), (105), (211), and (204) planes, respectively. As the calcination temperature increases, other peaks related to the $TiO_2$ rutile phase dramatically appear. CT-600 exhibits not only the sharp anatase peak, but also new rutile peaks at $2\theta$ = (27.44°, 36.12°, 41.2°, 44°, 54.32°, and 56.6°), corresponding to the (110), (101), (111), (210), (211), and (220) planes, respectively. When the sample is calcined at ever higher temperatures of 700 °C (CT-700), the dominant rutile peaks become even sharper. As the calcination temperature increases further to 800 °C, major rutile peaks with a small trace of the anatase peak are observed. Based on the above results, it should be noted that the metastable anatase phase was continuously converted to the rutile phase by the thermal transformation of the $TiO_2$ crystalline phase.

The average composition of anatase to rutile phase was calculated from the relative peak area of anatase (101) and rutile (110) peaks, using the following equation [43]:

$$[A]/\% = 100 \times I_A/(I_A + 1.265 \times I_R) \tag{1}$$

where $I_A$ and $I_R$ correspond to the relative areas of the anatase (101) and rutile (110) peaks, respectively. Hence, the rutile content is $[R] = 100 - [A]$. The rutile contents of the CT-X samples were estimated to be approximately (0%, 67%, 88%, and 100%) for the CT-500, CT-600, CT-700, and CT-800, respectively.

Figure 3b also shows the XRD patterns of the reduced $TiO_2$ sample. RT-500 sample shows the anatase diffraction peaks that are identical XRD patterns to the CT-500 sample. This indicates that the crystalline properties of the RT-500 sample are well maintained, even after the chemical reduction process using Li/EDA solution. RT-600 sample shows similar mixed crystalline patterns of both the anatase phase and rutile phase, but it shows a slightly higher relative peak intensity of anatase, compared to that of CT-600. As the calcination temperature increases, the anatase peak intensity of RT-x samples is interestingly enhanced, compared to that of CT-x. RT-700 and RT-800 exhibit the more obvious anatase (101) peaks, compared to CT-700 and CT-800 samples. In practical terms, the rutile contents were calculated to be approximately (0%, 66%, 84%, and 93%) for the RT-500, RT-600, RT-700, and RT-800 samples, respectively. It should be noted that after Li/EDA reduction, the anatase ratio is obviously increased, while that of the rutile is slightly decreased. It was recently reported that the rutile phase could be selectively disordered by Li in superbase EDA solution [39]. Interestingly, Li/EDA solution as the reducing agent could reduce the ordered white rutile to the disordered black one, resulting in the colored $TiO_2$ with diminution of the rutile phase. In our study, we also observed similar phenomena by using our calcined $TiO_2$ samples, which can have either anatase or rutile phase. Since Li/EDA reducing solution can selectively disorder the surface of rutile phase in the calcined $TiO_2$ spherical particles, significant chemical reduction can induce a decrease of the rutile surface orderliness, resulting in both a decrease of the rutile peak intensity and an increase of the anatase peak, respectively. Thus, the relative content of the anatase phase of RT-x sample increases, compared to that of the CT-x one. However, because the Li/EDA solution cannot completely reduce all the rutile content, the majority of rutile phases on RT-600, RT-700, and RT-800 samples still remain, with obvious appearance of the anatase peaks.

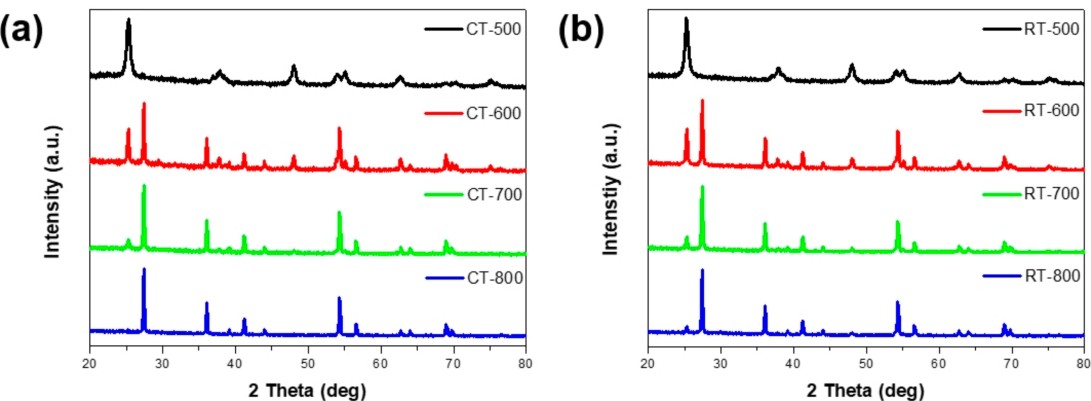

**Figure 3.** X-ray diffraction (XRD) patterns of (**a**) calcined $TiO_2$ sample (CT-x) and (**b**) reduced $TiO_2$ sample (RT-x).

The Raman spectra are also obtained to confirm the surface-structural changes of $TiO_2$ samples (Figure 4). As expected, the as-synthesized amorphous $TiO_2$ particle (AT) does not display any obvious peaks related to the crystalline structure. It showed small intensity changes at the Raman shift of ca. $1000 \text{ cm}^{-1}$, indicating the existence of a disordered surface [44]. CT-600 showed the obvious Raman peaks at Raman shift of ca. 400, 550, and $650 \text{ cm}^{-1}$, respectively, indicating representative ordered anatase characteristics. After the disordering process using the Li/EDA solution, RT-600 not only showed the similar Raman peaks at ca. 400, 550, and $650 \text{ cm}^{-1}$, but also obvious signal changes at ca. $1000 \text{ cm}^{-1}$. It should be noted that the disordered surface was formed by the Li/EDA reduction process. The RT-800 sample, which consists of mainly the rutile phase, displays a significant peak related to the disordered surface of rutile. The above features are also consistent with the trend of the XRD results. Based on the above XRD and Raman spectra, we conclude that the surface disordering of rutile can be easily achieved by a simple Li/EDA reduction process. Based on our observation, it can be considered that the original $Ti^{4+}$ state in the boundary of the rutile crystalline grain is preferentially reduced to $Ti^{3+}$ in superbase conditions in the presence of Ethylenediamine, which is a similar phenomenon to that previously reported [39].

We also confirmed if the disordered rutile surface could be recovered to the ordered surface by recalcining the RT-800 sample at 800 °C. As shown in Figure S1, CT-800 showed the obvious peaks at Raman shift of ca. 451 and $615 \text{ cm}^{-1}$, respectively, indicating the ordered rutile surfaces. After Li/EDA treatment, Raman peaks related with the ordered rutile surface of RT-800 were completely disappeared indicating the selective disordering of the rutile surface. When the sample is recalcined again at 800 °C, RT-800 recalcination exhibited the peaks of the ordered rutile surface again, indicating recovery of the disordered surface to the ordered counterpart. It should be noted that the ordered rutile surface can first be disordered through the Li/EDA reduction process, then the disordered surface can be recovered to the original ordered rutile surface by heat treatment. Based on our observation, it can be concluded that the original $Ti^{4+}$ state in the boundary of the rutile crystalline grain is reduced to $Ti^{3+}$, then re-oxidized to $Ti^{4+}$ states by sequential Li/EDA reduction and recalcination, respectively. It is consistent with the results of a previous study reported by Park et al. [39]. Since the disordered surface indicates different optical property, such as narrowed band-gap and enhanced absorption in visible light, it could be believed that our reduced $TiO_2$ sample showed different band-gap properties and enhanced photochemical performance under visible light conditions.

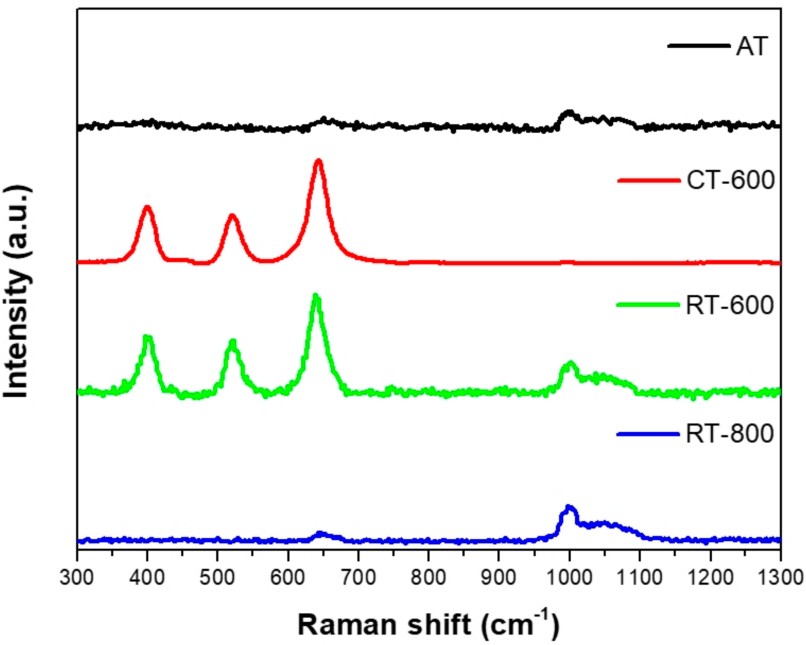

**Figure 4.** Raman spectra of amorphous TiO$_2$ (AT), CT-600, RT-600, and RT-800.

To investigate the light absorption ability and optical property of TiO$_2$ samples, we carried out UV-Vis diffuse reflectance spectroscopy (UV-Vis DRS). We obtained the absorption spectra of the TiO$_2$ samples employed in this work by UV-Vis DRS techniques (Figure 5). The as-synthesized amorphous TiO$_2$ sample (AT) can only absorb the UV region, indicating poor light absorbance towards solar light. While the calcined TiO$_2$ sample at 600 °C (CT-600) displays its main absorption of UV light until ca. 400 nm, the reduced TiO$_2$ sample (RT-600) shows the absorption edge red-shifted and significant increase of light absorption in the range ca. 400–650 nm, indicating large adsorption in the range of visible light. The RT-800 sample has even broader absorption in the range of visible light. We also estimated the band-gap energy of the above samples by using Tauc plot [45]. The band gap energy values of CT, CT-600, RT-600, and RT-800 are calculated as ca. 3.18, 2.93, 2.44, and 0.82 eV, respectively.

It is well known that pure TiO$_2$ shows a band-gap of ca. 3.0–3.2 eV [18]. In this work, the CT-600 sample, which has the mixed phase of anatase and rutile, displays a quite wide band-gap value of ca. 2.93 eV, even though there is small difference compared to previous results [10,11,18]. After chemical reduction using the Li/EDA solution, the RT-600 sample shows a narrowed band-gap (ca. 2.44 eV) with large absorption toward visible light. As previously reported, the Li/EDA solution can selectively make the disordered rutile surface of TiO$_2$ with a mixed phase of anatase and rutile [39]. Although undergoing the same Li/EDA reduction process, the anatase surface of the RT-500 sample can be well maintained, but some rutile surface of RT-600 can be selectively disordered. It is well known that the disordered surface can increase visible light absorption and utilize low-energy light on photocatalysis, which originates from either the regulated conduction band (CB) or valence band (VB) position, and indirect electron recombination [46]. In this study, we also observed that the reduced TiO$_2$ samples (RT-600 and RT-800) had enhanced light absorption ability and optical property, which should originate from the disordered surface of the rutile phase. Thus, during photocatalysis, our reduced TiO$_2$ samples should show enhanced performance.

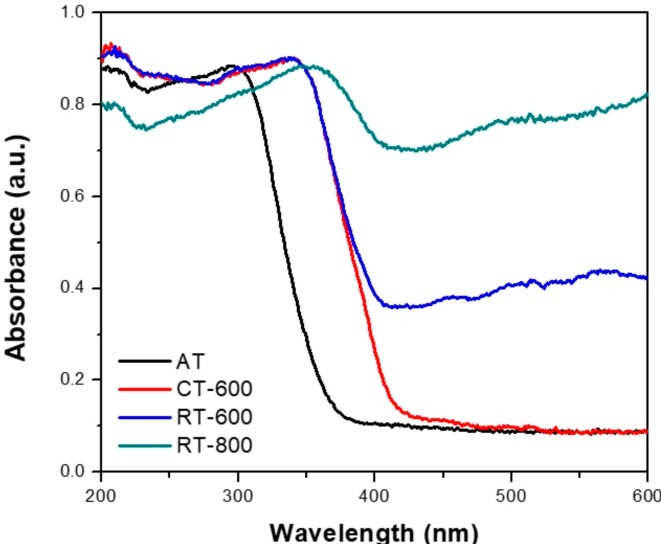

**Figure 5.** UV-Vis diffuse reflectance spectroscopy (UV-Vis DRS) spectra of amorphous $TiO_2$ (AT), CT-600, RT-600, and RT-800.

The photocatalytic hydrogen production activity of the $TiO_2$ sample was investigated under solar light (1.5 air mass, 1.5 AM) irradiation using methanol/$H_2O$ solution (Figure 6a,b). Before employing the synthesized $TiO_2$ samples as photocatalysts, 1 wt.% of Pt was deposited on the surface of each $TiO_2$ sample. Figure 6a shows that when the Pt/CT-600 sample was used as the photocatalyst, only a negligible amount of hydrogen was produced. This indicates the low photocatalytic activity of Pt/CT-600, which is attributed to small photon absorption of the CT-600 sample toward solar light, which consists of mainly visible and infrared light. However, Pt/RT-600 shows remarkable improvement in photocatalytic hydrogen production (Figure 6a). It should be noted that both the improved visible light absorption and the narrowed band-gap of RT-600 enhance the light absorption, resulting in the dramatically improved photocatalysis activity of Pt/RT-600. Figure 6b shows the effect of calcination temperature of RT-x samples on photocatalytic hydrogen production with light irradiation time. Among the catalysts tested, the Pt/RT-600 catalyst shows the highest catalytic activity. The relative photocatalytic activity of the catalysts toward hydrogen production follows the order: Pt/RT-600 > Pt/RT-500 > Pt/RT-700 ≈ PT/RT-800.

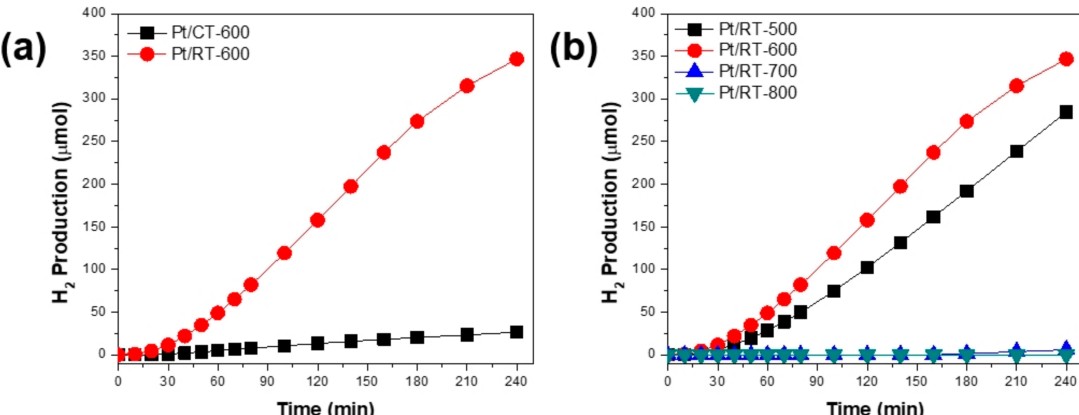

**Figure 6.** Hydrogen production amount of 1 wt % Pt-deposited $TiO_2$ photocatalyst: (**a**) CT-600 and RT-600; and (**b**) RT-500, RT-600, RT-700, and RT-800.

The photocatalytic activity of RT-x samples can be explained from the above characterization results. Since all RT-x samples was prepared from the same colloidal amorphous $TiO_2$ by calcination at different temperatures, followed by the same Li/EDA reduction process, it should be noted that the performance differences originate from the different degrees of rutile surface disordering, which is ascribed to different crystalline properties. The RT-500 sample, which is mainly anatase phase, has a small amount of disordered surface. In addition, since there is a small portion of UV-light in solar light, Pt/RT-500 can generate a considerable amount of hydrogen. The RT-600 sample, which consists of a mixed crystalline phase of anatase and rutile, can not only favorably absorb visible light over the reduced rutile, but it can also separate charge carriers to anatase, resulting in a lot of elongated charge carriers to accelerate photocatalysis. It is well known that the outstanding activity of P25-$TiO_2$ is mainly contributed by the mixed phase composition of anatase and rutile, which has beneficial effects on the absorption and separation of excited charges. Even though there is still controversy over the exact functions of each crystalline phase in the mixed phase, it is certain that the existence of the anatase–rutile mixed phase can have an unexpected and beneficial performance in photocatalysis [47,48]. Our RT-600 sample also had similar beneficial effects on light absorption and charge separation, due to the existence of the mixed phase. In practical terms, Pt/RT-600 shows the best hydrogen production performance. However, as the calcination temperature increases, the crystalline phase becomes mainly a rutile phase. RT-700 and RT-800 have a major rutile crystalline phase and a large portion of the disordered surface, resulting in the large absorption of visible light in the UV-DRS data. Although they can absorb visible light, the recombination of excited electron-hole pairs is severely constrained, due to the highly disordered surface and negligible portion of anatase. Although RT-800 sample can absorb the most visible light and the color of the catalyst is dark blue, Pt/RT-800 shows negligible hydrogen production.

To further confirm the photochemical properties of the $TiO_2$ samples, we measured the photocurrent by conducting chronoamperometry (CA) under an inducing potential of 0.6 V (vs. Ag/AgCl) and a periodic irradiation of solar light with a 400 nm cut-off filter. Figure 7 shows that the photocurrent is closely related to the degree of charge separation under light irradiation. Without light exposure, samples display electrochemical currents. When the catalysts are irradiated by light conditions, the current density can be increased due to the contribution from photo-generated electrons. The $TiO_2$ sample (CT-600) calcined at 600 °C exhibits a considerable photocurrent, indicating the existence of charge separation by light irradiation. The RT-600 sample shows the largest photocurrent among the $TiO_2$ samples tested. However, the RT-800 sample displays smaller photo-generated current than RT-600, indicating the restricted recombination of electron-holes, even though it can absorb a large portion of solar light. Based on both the characterization results and photoelectrochemical data, it can be concluded that the RT-600 photocatalyst, which consists of mixed phase with a disordered rutile surface, can have advantageous effects, such as visible light absorption, and favorable charge separation, resulting in improved photocatalysis activity on photocatalytic hydrogen production.

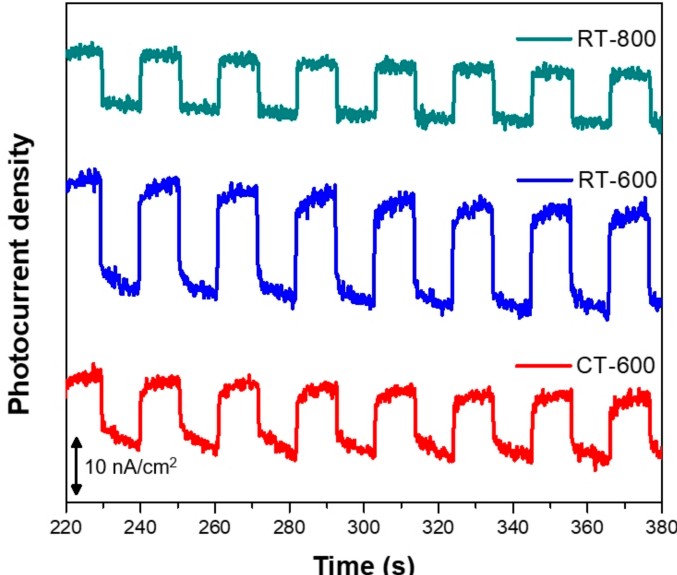

**Figure 7.** Photoelectrochemical chronoamperometry curves of various TiO$_2$ samples obtained at 0.6 V vs. Ag/AgCl under solar light illumination with 400 nm cut-off filter.

## 3. Materials and Methods

### 3.1. Materials

Ethyl alcohol (C$_2$H$_5$OH, 99.9%, anhydrous), Methyl alcohol (CH$_3$OH, 99.9%, anhydrous), acetonitrile (ACN, CH$_3$CN 99.9%, special guaranteed grade), and ammonium hydroxide (NH$_4$OH, 28%) were obtained from Daejung Chemical Company. Titanium (IV) n-butoxide (TBOT, 97%, reagent grade) and Hydroxypropyl cellulose (HPC, MW $\approx$ 80,000) were obtained from Aldrich. Metallic Li foil was purchased from Alfa Aesar chemical company. Ethylenediamine (EDA, 99%, guaranteed grade) was obtained from Sigma-Aldrich. All chemicals were used as received.

### 3.2. Synthesis

Spherical TiO$_2$ particles with average diameter of 290 nm were prepared by a sol-gel synthesis in mixed solvent, followed by calcination, as recently reported [42]. HPC (50 mg), as a surfactant, was completely dissolved in the mixed solvent solution (100 mL) of ethanol and acetonitrile with a volume ratio of 3:1. After completely dissolving the HPC, ammonia solution (0.8 mL) was added to the above solution. After stirring for 20 min, a mixture of TBOT (4 mL) in the mixed solvent of ethanol (12 mL) and acetonitrile (4 mL) was quickly injected into the above solution. The mixture was vigorously stirred for 2 h at room temperature. The white precipitate was isolated by centrifugation, and washed with ethanol and with de-ionized (D.I.) water several times. Then, the amorphous AT sample was obtained by drying under vacuum.

The dried AT sample was charged in an alumina boat in a furnace, and calcined at the desired temperature for 3 h under air conditions. The amorphous AT samples were crystallized to either anatase or rutile phase, which is highly dependent on the calcination temperature. Calcined CT samples are denoted as CT-X (where X is the calcination temperature). To synthesize colored TiO$_2$ spherical particle, we used the Li-EDA reduction method, which was recently developed by Park et al. [39]. A piece of metallic Li foil (45 mg) was dissolved in ethylenediamine (40 mL) to form a solvated electron solution. The calcined CT-X (400 mg) samples were added into the above solution, and vigorously stirred for 6 days under inert (N$_2$) conditions. After 6 days, a diluted HCl solution (1 M, 6.5 mL) was slowly added dropwise into the mixture, in order to quench the excess electron.

The reduced $TiO_2$ particles were isolated by centrifugation, washed with D.I. water 3 times, and dried in vacuum chamber at room temperature to give RT-X (where X is the calcination temperature).

### 3.3. Characterization

Digital photo images of $TiO_2$ samples were obtained by the digital camera function of iPhone (Apple Inc.). The particle morphology and uniformity were investigated by scanning electron microcopy (SEM, JSM-6060, JEOL). The crystalline properties of the $TiO_2$ samples were investigated by X-ray diffraction (XRD, D/MAX 2200, Rigaku). Optical absorbance spectra were studied by UV-vis spectroscopy using a UV-vis spectrophotometer with diffuse reflectance accessary (UV-DRS, V670, Jasco). Photoelectrochemical analyses were carried out using a conventional three-electrode system, with Ag/AgCl as a reference electrode, and Pt gauze as a counter electrode. The working electrode was prepared by deposition of a sample slurry on indium-tin oxide (ITO) glass ($1 \times 1$ cm) [12]. An aqueous $Na_2SO_4$ (0.1 mol/L) solution containing methanol (10 vol.%) was used as the electrolyte. Chronoamperometry tests were conducted using a potentiostat (SP-150, BioLogic).

### 3.4. Photocatalytic Hydrogen Production

Photocatalytic hydrogen production was conducted in a Pyrex glass reactor. $TiO_2$ photocatalyst samples (20 mg) were well dispersed in an aqueous methanol solution (50%, 50 mL). An ABET 150 W Xe lamp (ABET technologies inc. USA) with an AM 1.5 G air mass filter was used as a light source for solar light irradiation. The amount of hydrogen produced was determined by conventional gas chromatography with a thermal conductivity detector (GC-TCD, HP-5890 equipped with a Molecular Sieve-5A packed column).

## 4. Conclusions

We synthesized the uniform spherical $TiO_2$ particles with disordered rutile surface, characterized both the physicochemical and optical properties, and demonstrated photocatalytic performances on photochemical hydrogen production under solar light conditions. The synthesis involves several sequential processes: (i) Synthesis of uniform-sized amorphous $TiO_2$ particles by sol-gel reaction of the $TiO_2$ precursor, (ii) calcination of amorphous $TiO_2$ sample to convert to its crystalline counterpart, and (iii) chemical reduction of the calcined $TiO_2$ sample to make the disordered rutile surface. The as-synthesized amorphous $TiO_2$ sample showed uniform and monodispersed spherical morphologies. Calcination at varied temperatures induced different crystalline characteristics, such as different ratios of anatase and rutile phases, and chemical reduction using Li/EDA enables selective disordering of the rutile surface, resulting in different optical characteristics, such as the degree of visible light absorption ability and band-gap energy. The chemically-reduced $TiO_2$ sample (RT-600) prepared by calcination at 600 °C, followed by Li/EDA reduction, displays beneficial characteristics in terms of light absorption and charge separation, such as a disordered rutile surface, and a mixed crystalline phase of anatase and rutile. Among the $TiO_2$ samples employed in this work, the RT-600 sample showed significantly enhanced catalytic activity in photocatalytic hydrogen production. We believe that the proposed technique reported in this study can provide an effective method for developing visible light-responsive $TiO_2$-based photocatalysts.

**Supplementary Materials:** The following are available online at http://www.mdpi.com/2073-4344/9/6/491/s1, Figure S1: Raman spectra of CT-800, RT-800, and RT-800 recalcination.

**Author Contributions:** Investigation, writing—original draft, N.Y.K.; investigation, H.K.L.; resources, J.T.M.; conceptualization, writing—review and editing, J.B.J.

**Funding:** This research was funded by Konkuk University in 2016.

**Conflicts of Interest:** The authors declare no conflicts of interest.

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
