# Peer review of "Synthesis of Spherical TiO2 Particles with Disordered Rutile Surface for Photocatalytic Hydrogen Production"

_catalysts, doi:10.3390/catal9060491_

Round 1

Reviewer 1 Report

This work presented a method to make spherical TiO2 with disordered rutile surface. The obtained TiO2 has high visible light response toward photocatalytic H2 generation. The work is interesting considering the selective disorder on rutile phase by treating with Li/EDA. 

Several questions the authors still need to address on: 

By treating with Li/EDA, how to know the ratio of disordered rutile phase? If the disordered rutile distributed on the surface, what is the possible depth profile? 

Why the disordered rutile phase does not recover from high temperature (such as 800oC) treatment? The defects should be related with Ti3+ state, it could be oxidized and recovered o Ti4+ in the crystalline structure at elevated temperature treatment. Please explain. 

In Raman spectra(Fig. 4), why the peaks assigned to rutile is not presented even it is dominated. 

It can be accepted upon answering above questions 

Author Response

Response to comments from Reviewer #1

We appreciate valuable comments from the reviewer. Revisions were made according to the reviewer’s comments. Questions raised by the reviewers were answered point by point as follows. The revised parts were marked in red in the revised manuscript.

Reviewer #1: This work presented a method to make spherical TiO2 with disordered rutile surface. The obtained TiO2 has high visible light response toward photocatalytic H2 generation. The work is interesting considering the selective disorder on rutile phase by treating with Li/EDA. Several questions the authors still need to address on:

Question 1: By treating with Li/EDA, how to know the ratio of disordered rutile phase? If the disordered rutile distributed on the surface, what is the possible depth profile?

Answer: We thank the Reviewer for valuable comments. As pointed out by the Reviewer, it is important to quantify how much rutile phase is changed to the disordered counterpart. Although exact quantification of the disordered rutile phase is important, we would like to emphasize intensively the systemic control of crystalline properties and the resulted optical properties of the colored TiO2 sample after Li/EDA reduction with keeping other physicochemical characteristics. In addition, this work is one portion of our systemic study on colored TiO2 research. Currently, we are studying on controlling degree of the disordered surface of rutile based-TiO2 samples and relationship between degree of the disordered surface and performance enhancement in photocatalysis. The results are quite promising, and we will report the above results and discussions on our next publications soon.

Question 2: Why the disordered rutile phase does not recover from high temperature (such as 800 oC) treatment? The defects should be related with Ti3+ state, it could be oxidized and recovered to Ti4+ in the crystalline structure at elevated temperature treatment. Please explain.

Answer: We thank the Reviewer for valuable comments. As indicated in synthesis section, we first synthesized amorphous TiO2 particle and calcined it at desired temperature to get crystalline TiO2 sample (CT-X). Calcined CT samples are treated with Li-EDA solution resulting in reduced TiO2 sample (RT-X). These RT-X sample directly used as photocatalysts after deposition of Pt nanoparticle for photocatalytic hydrogen production. We did not do further heat-treatment after Li-EDA reduction step. In order to avoid confusion, we delete the sentence “(where X is the calcination temperature)” in synthesis section in Materials and Methods.

In addition, as the Reviewer commented, we confirmed if the disordered rutile surface can be recovered to the ordered surface by recalcining the RT-800 sample at 800 oC. As shown in Figure S1, CT-800 showed the obvious peaks at Raman shift of ca. (451, and 615) cm-1, respectively, indicating the ordered rutile surfaces. After Li/EDA treatment, Raman peaks related with the ordered rutile surface of RT-800 were completely disappeared indicating the selective disordering of the rutile surface. When the sample is recalcined again at 800 oC, RT-800-Recalcination exhibited the peaks of the ordered rutile surface again, indicating recovery of the disordered surface to the ordered counterpart. It should be noted that the ordered rutile surface can first be disordered through the Li/EDA reduction process, then the disordered surface can be recovered to the original ordered Rutile surface by heat-treatment. Based on our observation, it can be concluded that original Ti4+ state in the boundary of rutile crystalline grain is reduced to Ti3+, then re-oxidized to Ti4+ states by sequential Li/EDA reduction and recalcination, respectively. It is consistent with the results of previous study reported by Park et al.

Figure S1. Raman spectra of CT-800, RT-800 and RT-800-recalcination.

According to the Reviewer comments, we revised the manuscript. We added the following sentences and paragraph in the results and discussion section.

We also confirmed if the disordered rutile surface can be recovered to the ordered surface by recalcining the RT-800 sample at 800 oC. As shown in Figure S1, CT-800 showed the obvious peaks at Raman shift of ca. (451, and 615) cm-1, respectively, indicating the ordered rutile surfaces. After Li/EDA treatment, Raman peaks related with the ordered rutile surface of RT-800 were completely disappeared indicating the selective disordering of the rutile surface. When the sample is recalcined again at 800 oC, RT-800-Recalcination exhibited the peaks of the ordered rutile surface again, indicating recovery of the disordered surface to the ordered counterpart. It should be noted that the ordered rutile surface can first be disordered through the Li/EDA reduction process, then the disordered surface can be recovered to the original ordered Rutile surface by heat-treatment. Based on our observation, it can be concluded that original Ti4+ state in the boundary of rutile crystalline grain is reduced to Ti3+, then re-oxidized to Ti4+ states by sequential Li/EDA reduction and recalcination, respectively. It is consistent with the results of previous study reported by Park et al. [39].”

[39] Zhang, K.; Wang, L.; Kim, J.K.; Ma, M.; Veerappan, G.; Lee, C.-L.; Kong, K.-j.; Lee, H.; Park, J.H. An order/disorder/water junction system for highly efficient co-catalyst-free photocatalytic hydrogen generation. Energy & Environmental Science 2016, 9, 499-503

Question 3: In Raman spectra (Fig. 4), why the peaks assigned to rutile is not presented even it is dominated.

Answer: We thank the Reviewer for valuable comments. As the Reviewer pointed out, there is no obvious peak related to rutile phase of RT-x sample in the Raman spectra even though it is dominant in XRD analysis. Since the Raman spectrum is highly influenced by focus of laser beam and our purpose is to find disordering phenomenon by Li/EDA reduction process, Raman analysis are focused pretty much on surface analysis. We conducted the Raman spectroscopy analysis with focus on surface analysis. As shown in Figure S1, CT-800 exhibits strong Raman peaks related with ordered rutile surface and peaks related with the ordered rutile surface of RT-800 were completely disappeared by Li-EDA treatment. Based on the Raman and XRD results, it can be considered that bulk in the RT-800 is mainly consisted of the rutile orderness but surface is disordered. When the sample is recalcined at 800 oC, obvious Raman peaks related to rutile surface are observed, since the disordered surface is recovered to the ordered one. Based on the XRD and Raman results, it can be thought that the boundary surface of rutile crystal grain is preferentially disordered rather than bulk in particle, resulting that the reduced TiO2 has the disordered rutile surface with ordered rutile bulk in rutile crystalline grain.

Reviewer 2 Report

Authors reported a synthesis route of spherical TiO2 with controllable disorder rutile surface for hydrogen evolution. Overall, this work is significant and novel, one question is not clear:

Authors mentioned the chemical reduction by Li/EDA enables selective reduction of the rutile surface. As the reduction correlates with the disorder degree and crystal phase ratio,  the reduction (or selective reduction) mechanism is needed. 

Author Response

Response to comments from Reviewer #2

We appreciate valuable comments from the reviewer. Revisions were made according to the reviewer’s comments. Questions raised by the reviewers were answered point by point as follows. The revised parts were marked in red in the revised manuscript.

Reviewer #2: Authors reported a synthesis route of spherical TiO2 with controllable disorder rutile surface for hydrogen evolution. Overall, this work is significant and novel, one question is not clear:

Question 1: Authors mentioned the chemical reduction by Li/EDA enables selective reduction of the rutile surface. As the reduction correlates with the disorder degree and crystal phase ratio, the reduction (or selective reduction) mechanism is needed.

Answer: We thank the Reviewer for valuable comments. As pointed out by the Reviewer, it is important to quantify how much rutile phase is changed to the disordered counterpart. Although exact quantification of the disordered rutile phase is important, we would like to emphasize intensively the systemic control of crystalline properties and the resulted optical properties of the colored TiO2 sample after Li/EDA reduction with keeping other physicochemical characteristics. In addition, this work is one portion of our systemic study on colored TiO2 photocatalysts research. Currently, we are studying on controlling degree of the disordered surface of rutile based-TiO2 samples and relationship between degree of the disordered surface and performance enhancement in photocatalysis. The results are quite promising, and we will report the above results and discussions on our next publications soon

In point of reduction mechanism, based on our observation, it can be considered that original Ti4+ state in the boundary of rutile crystalline grain is preferentially reduced to Ti3+ in superbase conditions in the presence of ethylenediamine reported by Park et al. [39].

According to the Reviewer comments, we revised the manuscript. We added the following sentences and paragraph in the results and discussion section.

“Based on our observation, it can be considered that original Ti4+ state in the boundary of rutile crystalline grain is preferentially reduced to Ti3+ in superbase conditions in the presence of ethylenediamine, which is a similar phenomenon to that previously reported [39].”

[39] Zhang, K.; Wang, L.; Kim, J.K.; Ma, M.; Veerappan, G.; Lee, C.-L.; Kong, K.-j.; Lee, H.; Park, J.H. An order/disorder/water junction system for highly efficient co-catalyst-free photocatalytic hydrogen generation. Energy & Environmental Science 2016, 9, 499-503